# Identification of the Key Genes Associated with the Yak Hair Follicle Cycle

**DOI:** 10.3390/genes13010032

**Published:** 2021-12-23

**Authors:** Xiaolan Zhang, Pengjia Bao, Na Ye, Xuelan Zhou, Yongfeng Zhang, Chunnian Liang, Xian Guo, Min Chu, Jie Pei, Ping Yan

**Affiliations:** Key Laboratory of Yak Breeding Engineering Gansu Province, Lanzhou Institute of Husbandry and Pharmaceutical Sciences, Chinese Academy of Agricultural Sciences, Lanzhou 730050, China; zhangxl8997@163.com (X.Z.); baopengjia@caas.cn (P.B.); yena199306@163.com (N.Y.); zhouxl17@lzu.edu.cn (X.Z.); zhangyongfeng_ying@163.com (Y.Z.); chunnian2006@163.com (C.L.); guoxian@caas.cn (X.G.); chumin@caas.cn (M.C.); peijie@caas.cn (J.P.)

**Keywords:** hair follicles cycle, lncRNA-mRNA, WGCNA, hub genes, yak

## Abstract

The development of hair follicles in yak shows significant seasonal cycles. In our previous research, transcriptome data including mRNAs and lncRNAs in five stages during the yak hair follicles (HFs) cycle were detected, but their regulation network and the hub genes in different periods are yet to be explored. This study aimed to screen and identify the hub genes during yak HFs cycle by constructing a mRNA-lncRNA co-expression network. A total of 5000 differently expressed mRNA (DEMs) and 729 differently expressed long noncoding RNA (DELs) were used to construct the co-expression network, based on weighted genes co-expression network analysis (WGCNA). Four temporally specific modules were considered to be significantly associated with the HFs cycle of yak. Kyoto Encyclopedia of Genes and Genomes (KEGG) analysis revealed that the modules are enriched into Wnt, EMC-receptor interaction, PI3K-Akt, focal adhesion pathways, and so on. The hub genes, such as *FER*, *ELMO1*, *PCOLCE*, and *HOXC13*, were screened in different modules. Five hub genes (*WNT5A*, *HOXC13*, *DLX3*, *FOXN1*, and *OVOL1*) and part of key lncRNAs were identified for specific expression in skin tissue. Furthermore, immunofluorescence staining and Western blotting results showed that the expression location and abundance of DLX3 and OVOL1 are changed following the process of the HFs cycle, which further demonstrated that these two hub genes may play important roles in HFs development.

## 1. Introduction

The yak (*Bos grunniens*) is a key and symbolic species on the Tibetan Plateau. To resist the arctic–alpine and hypoxic environments, in addition to being covered with coarse wool in yak skin, a layer of undercoat (yak cashmere) grows on the bottom. Yak cashmere is a valuable textile material with fine fibers, similar to cashmere, and it has the characteristics of soft texture and warm performance, compared with coarse wool. This double-coated structure may be the result of yak adapting to the alpine environment for a long time. Coarse wool is derived from primary hair follicles (PHFs) and cashmere is derived from secondary hair follicles (SHFs). In yak, the scapular region, back, and side are prolific with cashmere, the abdomen is mainly coarse hair. The hair of yak is important for pastoralist’s living materials and economic benefit in the Tibetan Plateau. Similar to cashmere goats, the SHFs of yak also undergo a clear seasonal circulation. A growth cycle mainly consists of growth (anagen), regression (catagen), and a rest stage (telogen) [1]. 

The hair follicles maintain its normal periodic activity is the result of response to multiple signaling molecules and their complex interactions, these signals may differently express in stages or specifically express in skin tissue. Over the past several decades, a large of signals involving in HFs development have been revealed and studied extensively. Several important cytokines, such as IGF1, EGF, PDGF, hepatocyte growth factor, and VEGF, have been shown to control the maintenance in anagen of HFs [2,3]. *FGF5*, *BDNF*, *p53*, *TGFβ1*, and *BMPR1a* were identified to promote the induction of catagen [4,5,6,7]. In addition, transcription factors, such as MSX2/FOXN1 [8], HOXC13 [9], LHX2 [10], GATA3 [11], DLX3 [12], and LEF1 [13], were reported playing critical roles in hair follicle stem cell (HFSCs) activation, differentiation, and self-renewal by modulating the key pathways associated with hair follicle development. The pathways Wnt, BMP, Shh, Notch, and MAPK have been indicated as being involved in the regulation of hair cycle [14,15,16]. Anagen onset, for instance, is induced by an elevation in Wnts and the inhibition of BMP [17]. 

Long non-coding RNAs (lncRNAs) are defined as the transcripts of greater than 200 nt in length and no protein encoding ability. In recent years, lncRNAs have been reported as participating in various biological processes, including epigenetic regulation, cell cycle, cell differentiation, and proliferation, and were identified as vital regulators playing roles in the regulation of the chromatin structure or modulating the activity of proteins, mRNAs, or miRNAs [18]. The significantly seasonal change of the HFs in yak is important for its adaptability to alpine environment, which also makes yak an excellent model for the study of HFs cycle. In our previous works, transcriptome data, including mRNA and lncRNA, were measured in yak scapular skin at five time points (Jan., Mar., Jun., Aug., Oct.) during the HFs cycle, and the transcription characteristics and expression were, respectively, analyzed [19,20]. These five time points represent the critical stages during yak hair follicle cycling, among which, Jan. and Mar. are catagen and telogen, respectively, Aug. and Oct. are anagen, and Jun. is the transitional period from telogen to anagen. The regulatory genes and molecular interactions may have changed at different stages. Here, the mRNA-lncRNA co-expression network were constructed to further analyze, excavate, and explicit the molecular regulatory relationship and hub genes during yak hair cycle, on the basis of our previously elaborated studies. 

Weighted gene co-expression network analysis (WGCNA) is a widely used system for biological analysis. Based on the correlation between changes in gene expression signal values, it can be used to find highly correlated gene clusters (modules). Associating modules with external sample features or clinical character, and select hub genes within key modules, could be used as biomarkers or therapeutic targets in medicine [21]. Studies using WGCNA methods revealed the gene modules, signaling pathways, and hub genes that are associated with bovine fat deposition and mastitis development [22,23]. In goat, temporally specific modules and hub lncRNAs associated with goat skeletal muscle development were identified [24]. Compared with differential expression analysis, WGCNA can be used to better analyze the changes of overall biological processes, and the application in the study of dynamic changes of hair follicle development will be helpful in excavating more biological information related to the hair cycle.

## 2. Materials and Methods

### 2.1. Sample Collection

Yak tissues were used in tissue-specific detection, including skin, small intestine, heart, liver, spleen, lung, kidney, subcutaneous fat, muscle, the testis, which were collected after slaughter and immediately frozen in liquid nitrogen for further processing. There were three replicates in each HFs development stage and 15 RNA samples were used for RT-qPCR to verify the expression abundance of hub genes at different developmental stages of hair follicles in yak. The RNA samples were obtained from our laboratory from a previous experiment [19]. Skin tissues used in immunofluorescence and WB were collected in Jan., Apr., and Sep., according to the previous method [19]. All yaks in this study were from the Tianzhu white yak propagation bases of Wuwei City, Gansu Province of China. The experimental procedures were approved by the Animal Care and Use Committees of the Lanzhou Institute of Animal Science and Veterinary Medicine, the Chinese Academy of Agricultural Sciences.

### 2.2. Data Information and Data Selection

The data in this study used previous preliminary transcriptome sequencing results [19,20]. Additionally, the data also can be obtained from the public dataset PRJNA550233 in the National Center for Biotechnology Information (NCBI) BioProject (https://www.ncbi.nlm.nih.gov/bioproject/PRJNA550233, accessed on 20 October 2020). PRJNA550233 includes 15 yak skin samples from 5 developmental time points during the yak hair cycle (Jan., Mar., Jun., Aug., Oct.), with three repetitions in each time point. The PRJNA550233 dataset were generated using the HiSeq 2500 Illumina sequencing platform. In the present study, the FPKM data of mRNAs and lncRNAs at five time points in follicular development were used for pairwise comparisons, and the genes with significant differences in at least one comparison group can be used as differentially expressed mRNAs (DEMs) or differentially expressed lncRNAs (DELs). TheDEMs and DELs were screened with the criteria of *p*-value < 0.05 and |log2 (FPKMtime1/FPKMtime2)| ≥ 1. The comparison results of the DEMs and DELs are listed in Appendix A and Appendix A, respectively, which were used for WGCNA analysis. 

### 2.3. Co-expression Network Construction and KEGG Enrichment Analysis

The R package WGCNA V1.64.1 was used to construct lncRNA-mRNA co-expression network based on the FPKM data of DEMs and DELs [21]. A signed weighted correlation network was built by calculating gene expression similarity to create an adjacency matrix. In this study, we selected a soft threshold power, β = 9, to establish the co-expression network, so that the correlation between genes conformed to the scale-free network distribution. One step network module was constructed by merging genes with highly similar co-expression patterns into modules. The dynamic tree cut algorithm—with parameters set as: minModeSize = 30, mergeCutHeight = 0.25—was employed to cut the hierarchal clustering tree. Gene modules were distinguished as different colors, and were indicated by different branches on the clustering tree. The eigengenes of each module was determined. To identify temporal specific modules, the correlations of these modules with the different developmental stages in the yak hair cycle were investigated using WGCNA package. The module eigengenes closely related to specific time points were screened as temporally specific modules. Protein-coding genes of each key module were submitted to DAVID for KEGG pathway enrichment analysis. *Bos mutus* was used as a background list in DAVID. The default parameters of the number of genes greater than 2 and *p* value < 0.1 were used as thresholds to output the KEGG pathway enrichment results of different modules. Top 20 KEGG pathways of different modules were visualized using ggplot2 R package. 

### 2.4. Analysis of Temporally Specific Co-expression Network 

The co-expression networks closely related to distinct stages were, respectively, analyzed and visualized by Cytoscape (version 3.5.1). Cytohubba was applied to calculate the topological parameters of the nodes and screen the key nodes identified as hub genes. Meanwhile, protein–protein interaction (PPI) network analyses were used to analyze the interaction of interested eigengenes into the STRING database (https://string-db.org, accessed on 6 June 2021). The medium confidence of the PPI network was set as the interaction score > 0.4 and visualized by Cytoscape.

### 2.5. RNA Extraction, Real-Time Quantitative PCR, and PCR Detection

Total RNA of the tissues was isolated using the Trizol reagent (Invitrogen, Wilmington, NC, USA). The RNA concentration and quality were evaluated using a NanoDrop spectrophotometer (Thermo Scientific, Wilmington, DE, USA). The first-strand cDNA was synthesized using the PrimeScriptTM RT reagent kit (Takara, Dalian, China) according to the manufacturer’s instructions. The RT-qPCR was performed using TB GreenTM Premix Ex Taq II (Takara, Dalian, China), and the following reaction conditions were used: 95 °C for 3 min, followed by 40 cycles of 95 °C for 10 s, and 60 °C for 30 s. The relative expression of genes was normalized using *GAPDH* and analyzed with the 2^−ΔΔCt^ method. PCR amplification was performed using the 2 × EasyTaq PCR SuperMix (TransGen Biotech, Beijing, China). The PCR procedure was as follows: 95 °C for 3 min, followed by 35 cycles of 95 °C for 30 s, 60 °C for 30 s, and 72 °C for 20 s, and a final extension step was 72 °C for 5 min. *β-actin* and *GAPDH* were amplificated as two internal references. PCR amplification products were detected by 1.5% agarose gel. The primers used for RT-qPCR and PCR are listed in Appendix A and designed with oligo 6.

### 2.6. Histology and Immunofluorescence

Skin samples in different stages were sliced into 7 µm sections with a freezing microtome (Leica, Wetzlar, Germany), then, the sections were stained with primary antibodies overnight at 4 °C for immunofluorescent staining. The primary antibodies and dilutions used were anti-DLX3 (PA5-101065; ThermoFisher, USA, 1:100) and anti-OVOL1 (PA5-41480; ThermoFisher, USA, 1:100). Subsequently, secondary antibodies—Alexa Fluor^®^ 488 goat anti-rabbit (ab15007; abcam, 1:500)—were used to incubate against the specifical primary antibody. DAPI (Solarbio, Beijing, China) was used for nuclei staining. The slides were examined using fluorescence microscope (Leica, Germany).

### 2.7. Western Blotting

Total protein of yak skin tissues was extracted with RIPA Lysis Buffer containing protease inhibitor and protein phosphatase (Beyotime, Shanghai, China), and protein concentrations were determined by BCA Protein Assay Kit (Beyotime, Shanghai, China). The protein expression levels of DLX3 and OVOL1 in different developmental stages were quantified using a capillary-based “WES” Simple Western System (ProteinSimple, San Jose, CA, USA), according to the manufacturer’s instructions. The primary antibody information was as follows: anti-OVOL1 (PA5-41480; ThermoFisher; 1:50), anti-DLX3 (PA5-101065; ThermoFisher; 1:50), and GAPDH (10494-1-AP, Proteintech; 1:1000). The relative protein expression was calculated and analyzed based on the gel-like images produced by the Compass for SW software (Version 4.0, Protein Simple, SanJose, CA, USA). 

### 2.8. Statistical Analysis

All the experiments were repeated three times. The statistical analyses were performed by GraphPad Prism 8.0 software and the data were presented as the mean ± SEM. One-way ANOVA and the Dunnett’s test were used for multiple comparisons. *p* < 0.05 was considered to be statistically significant.

## 3. Results

### 3.1. Construction of mRNA-lncRNA Co-expression Network

Total 5000 differently expressed mRNAs (DEMs) (Appendix A) and 728 differently expressed lncRNAs (DELs) (Appendix A) were identified and used to construct the mRNA-lncRNA co-expression network by WGCNA software package in R. Nine co-expression modules were established (Figure 1a), excluding the genes in grey module that cannot be assigned to co-expression module. Number of the genes in different modules varied widely, from 45 genes in the magenta module to 1599 in turquoise module (Appendix A). An eigengene adjacency heatmap was generated to explore the correlations between the modules and the dendrogram branches with positively correlated eigengenes were grouped together (Figure 1b). Then, the heatmap was established for the correlation analysis between the stages of HFs development and module eigengenes to identify the temporally specific modules. As shown in Figure 1c, each stage was related to a module that has a high correlation with it. The yellow module has a high correlation with Jan. (*cor* = 0.85, *p* = 4 × 10^−5^); the turquoise module is closely associated with Mar. (*cor* = 0.74, *p* = 0.002); the red module is closely associated with Jun. (*cor* = 0.79, *p* = 5 × 10^−4^); the blue module is significantly associated with Oct. (*cor* = 0.69, *p* = 0.004). Therefore, the four modules were screened as temporally specific modules for the further functional analysis. Due to the black module closely associated with Aug., containing a smaller number of genes that failed in the functional enrichment analysis, Aug. was excluded.

### 3.2. Function Enrichment Analysis 

To understand the biological functions of the temporally specific co-expression modules, KEGG enrichment analysis was performed by submitting the mRNAs in key modules to DAVID database (Appendix A). KEGG enrichment results showed that the pathways, such as Basal cell carcinoma, Melanogenesis, Tight junction, and Wnt signaling pathway, were enriched in the blue module that related to Oct. (anagen), these pathways were reported to highly express anagen or involve induction of anagen hair follicle differentiation. The TNF signaling pathway, platelet activation, ECM–receptor interaction and PI3K-Akt signaling pathway, the pathways that involved in promoting hair follicle neogenesis, and those inducing folliculogenesis, were enriched in turquoise and red modules, the two modules are closely associated with Mar. and Jun., respectively. Mar. is in the telogen, and Jun. is in the transition of telogen to anagen during HFs cycle. In the yellow module, the cAMP signaling pathway, the MAPK signaling pathway, and the Ras signaling pathway were enriched, which were reported as involving in the induction of early catagen transition or hair loss. Furthermore, focal adhesion was enriched in several modules, including red, turquoise, and blue, implying that the pathway may play an essential role in hair follicles development (Figure 2a). Meanwhile, the scatterplot of gene significance (GS) for each period vs. module membership (MM) of their associated modules were plotted (Figure 2b). The results showed that all the GS and MM exhibit very significant correlation, suggesting that hub genes in the key modules also tend to be highly correlated with their closely related stages, which also demonstrated the reliable biological significance of these co-expression modules. 

### 3.3. Screening of The Hub Genes in Temporally Specific Modules 

To further analyze the potential regulatory relationship and screening the hub genes in the co-expression modules, the co-expression mRNA- lncRNA networks of the yellow, turquoise, red, and blues modules were analyzed, respectively. Genes of the top 100 degree of connectively in the different module networks were calculated and visualized using Cytohubba tab in Cytoscape. In the co-expression networks, the larger the node is, the higher connectivity in the network can be regarded as a potential hub regulator (Figure 3). The results showed that *FER*, *UGGT1*, *BDP1,* and *MED13* were identified to be the hub genes in the yellow module (Figure 3a). *ELMO1*, *CPEB1*, and *PID1* were located the hub position in the turquoise network (Figure 3b). *KDELR3* and *PCOLCE* were the hub genes in the red module (Figure 3c). In the co-expression network of the blue module, multiple transcription factors, including *HOXC13*, *FOXN1*, *MSX2*, *CUX1*, *DLX3*, and *OVOL1*, and keratins, such as *KRT35*, *KRT32*, and *KRT82*, which were the crucial regulators or structure proteins for the HFs development, were revealed, and had a higher connectivity in the networks (Figure 3d). All the lncRNAs were at the relative lower degree in the networks. In general, this section screened and visualized the hub genes in different period of HFs development in yak.

### 3.4. PPI Analysis and Expression Characteristics Detection of The Hub Genes in the Blue Module

Blue module was closely related to Oct. (Figure 1c), a period of full anagen during yak HFs cycle. In the blue module network, it was found that multiple hub genes were reported as being involved in HFs development. To further investigate the relationship between these hub regulators, PPI analysis was performed of the blue network. A subnetwork consisting of a series of transcription factors, such as HOXC13, MSX2, LEF1, DLX3, and FOXN1, and keratins including KRT32, KRT35, and KRT82, that were involved in HFs development, were presented in the PPI network (Figure 4a), which further indicates that these key genes may interact synergistically, promoting the hair growth. Then, the tissue specificity of part of the hub genes in the blue network and the PPI subnetwork were detected by PCR method. PCR result showed that, in addition to keratins, the expressions of *WNT5A*, *HOXC13*, *DLX3*, *FOXN1*, and *OVOL1* were also relatively specific in their expressions in skin (Figure 4b); these five genes also have been reported to be crucial regulators in HFs development. Then, the Pearson correlation coefficient between mRNAs and DELs were analyzed (Appendix A), the DELs with a Pearson correlation coefficient (r) > 0.9 or −0.9 and *p* value < 0.05 with the five key regulators were visualized. There were 27 DELs that were found closely associated with the five regulators, among which, four DELs were negatively correlated, and others were positively correlated (Figure 4c). The expression of lncRNAs were reported to be spatially and temporally specific [25]. Six lncRNAs were randomly selected for tissue specific detection, and the result showed that part of the lncRNAs relatively specific expressed in skin (Figure 4b). Moreover, the expression changes of *WNT5A*, *HOXC13*, *DLX3*, *FOXN1*, and *OVOL1* in different stages of HFs cycle were detected using RT-qPCR, which could also represent the expression trend of the genes in the blue module. The result showed that the five genes were highly expressed in anagen (Aug. and Oct.) (Figure 4d), consisting with the general expression trend of the blue module, analyzed by WGCNA. These results further identified and characterized the hub genes in blue, and analyzed the lncRNAs which interacted with key mRNAs. 

### 3.5. Spatial–Temporal Expression Analysis of DLX3 and OVOL1 during Yak HFs Cycle 

In order to further identify the hub genes that may play crucial roles in HFs development, *DLX3* and *OVOL1* were selected to analyze the spatiotemporal expression in protein level during the HFs cycle. *DLX3* and *OVOL1* were found located a core position in the blue co-expression network and their roles in HFs development of yak have been poorly studied. Immunofluorescent staining results showed that the expression abundance and location of DLX3 and OVOL1 in protein level changed during yak hair cycle. DLX3 expression was detected in the outer root sheath throughout the hair follicle cycle. Besides, by anagen stage DLX3 was specifically expressed in matrix and hair shaft, and was presented in epidermis chain of catagen. In telogen, DLX3 is highly expressed in the telogen bulge (Figure 5a). The result suggested that DLX3 may play a role in the whole process of the HFs cycle, a phenomenon which could also be reflected in Figure 5c. OVOL1 expression was detected in the hair matrix in anagen, and is mainly presented in the inner root sheath of the lower part of hair follicle in early anagen, full anagen, and later anagen. In catagen and telogen, the expression of OVOL1 is weaker, implicating that OVOL1 mainly regulates the growth of hair follicles in anagen (Figure 5b), and this expression distribution could also be observed in Figure 5c. Then, WB was used to determine the expression of DLX3 and OVOL1 in Jan. (catagen), Apr. (telogen), and Sep. (anagen), the result was basically consistent with the result of immunofluorescent. Both DLX3 and OVOL1 are highly expressed in Sep. (anagen). Compared with OVOL1, DLX3 has a higher expression abundance in each stage, and the expression of OVOL1 is lowest in Jan. (catagen) (Figure 5d).

## 4. Discussion

The hair cycle of mammals includes anagen, catagen, and telogen. The periodic transformation of the hair cycle is regulated by multiple signaling molecules and complex interactions between the epidermis and dermis. In animal husbandry, the normal hair cycle is of great significance to the economic benefit of fiber-producing animals, such as cashmere goat, sheep, and angora rabbit. The seasonal cycling of secondary HFs allows yak to have a thicker coat in cold winters and molt in warmer seasons, which are crucial for yak’s resistance to the alpine and harsh environments. In the present study, the differently expressed mRNAs and lncRNAs of yak skin at different stages during HFs cycling (Appendix A) was used to construct the mRNA-lncRNA co-expression network by WGCNA. There were four temporally specific modules that were found closely related to different HFs development stages, respectively (Figure 1c). These modules could be applied to screen and identify the hub genes and important signaling pathways of different periods of HFs cycle. KEGG analysis result showed that the signaling pathways enriched in the modules are consistent with the biological process of that module closely related period (Figure 2a). Wnt, basal cell carcinoma, melanogenesis, and tight junction were enriched in the blue module. Wnt has been considered as a key signaling pathway that drives HFs anagen induction and is required for hair follicle growth and regeneration [26,27]. Basal cell carcinoma, melanogenesis, and tight junction were reported to be highly expressed in anagen or as involving the induction of anagen hair follicle differentiation [28,29,30]. ECM–receptor interaction and PI3K-Akt signaling pathways were enriched in both the red and the turquoise modules. EMC plays an important role in the hair follicle stem cell niche-regulating, bulge cell behavior and hair follicle regeneration [31,32]. PI3K-Akt signaling is essential for de novo hair follicle regeneration [33]. Mar. is the telogen of yak HFs cycle; Jun. is the transitional period from telogen to anagen. During the stage of telogen to early anagen, the hair follicle stem cells transform from a quiescent to activated state, and this is a crucial stage of hair follicle neogenesis. Besides, platelet activation and TNF signaling pathways that involved in inducing folliculogenesis and promoting of hair follicle neogenesis were also enriched in turquoise and red modules [34,35]. In the yellow module, the cAMP signaling pathway [36], the MAPK signaling pathway [37], and the Ras signaling pathway [38] were reported to be involved in induction of early catagen transition or hair loss, the functions coincide with the biological process of HFs in Jan. (catagen). Thus, KEGG enrichment analysis revealed the biological functions of the temporally specific modules, as well indicated the validity of our module construction. 

In this study, hub genes of the key modules were screened. Genes located at the core of the network could be focused on as key regulatory genes. *Fer* is a hub gene located at the core of the yellow module network (Figure 3a), which was reported to be involved in the growth and metabolism of various epithelial tumors and was deemed to an important positive regulator of melanoma metastasis. A lack of Fer kinase in melanocytes emerged an in impaired activity of Wnt/β-catenin signaling pathway [39]. In the turquoise module (Figure 3b), the hub gene *ELMO1* was reported playing an unexpected role in the clearance of apoptotic cells and in the maintenance of homeostasis [40]; *CPEB1* inhibits epiderm–mesenchymal transformation and regulates cell cycles [41]. These functions of the hub genes may contribute to the maintenance and dynamics in telogen. *PCOLCE* and *KDELR3* are the prominent hub genes in red module associated with Jun. (Figure 3c). *PCOLCE* was reported to be an important factor in the transition from telogen to anagen [42], and the expression of *KDELR3* is associated with the development of melanocytes and the progression of melanoma [43]. Furthermore, several hub genes found in the blue module (Figure 3d), including *MSX2*, *FOXN1* [8], *HOXC13* [44], *DLX3* [12], *OVOL1* [45], *LEF1* [46], *TCHH*, and *PADI3* [47,48], are well known to be crucial regulators of hair follicle development and are located in the core of the network. In addition, more keratins, such as KRT82, KRT35, KRT35, KRT28, and KRT71, were showed in the blue module. The blue module was related to anagen (Oct.); anagen is the longest stage of the hair follicle development cycle, accounting for more than half of the whole growth cycle. The anagen in yak hair cycle is from about Jun. to Dec. Moreover, researchers are also likely to focus on growing hair follicle due to the intact morphology, which may lead to more genes involved in regulation of hair follicle development to be found in the blue module. The intuitive display of hub genes in different stages of the hair cycle would provide reference information for further mining of regulators related to the hair cycle and reveal the mechanism of hair follicle development. 

Multiple regulators and keratins in the blue co-expression network (Figure 3d) were also found to constitute a PPI network (Figure 4a), which further indicates that these genes may interact with each other in hair follicle development. Among these key genes, in addition to several keratins, the regulators *WNT5A*, *HOXC13*, *DLX3*, *FOXN1*, and *OVOL1* were also found to be specifically expressed in skin tissue (Figure 4b). The regulators *WNT5A*, *HOXC13*, and *FOXN1* have been widely reported to play important roles in hair follicle development [8,9,49]. 

*DLX3* and *OVOL1* were two hub genes in the blue module and were specifically expressed in skin. DLX3 is a transcription factor that regulates epidermal, neural, and osteogenic cellular differentiation [12]. OVOL1 is a putative transcription factor and is mainly involved in hair formation and spermatogenesis [50]. In the present research, DLX3 was shown to be expressed in the outer root sheath, throughout the hair follicle cycle, and expressed in matrix and hair shaft in anagen, and in the epidermis chain in catagen (Figure 5a). Detection of the expression position of DLX3 in our study is consistent with previously reported data [12]. Our results intuitively and globally demonstrated the expression of DLX3 in yak hair follicle. Changes of DLX3 in expression abundance and location during the HFs cycle further evidenced that DLX3 is a crucial regulator in hair morphogenesis, differentiation, and cycling programs. OVOL1 was detected, mainly presenting in the inner root sheath of the lower part of the hair follicle in the early anagen, mid-anagen, and later anagen, and was expressed in the hair follicle matrix in mid-anagen (Figure 5b); the result is consistent with the confirmed expression of OVOL1 in the inner root sheath of human hair follicle [45,51]. The OVOL1–OVOL2 axis was reported to be crucial for the normal development of hair follicles in murine. OVOL1-deficient mice indicate ruffled hair coat and hair abnormalities [50]. However, the role of *OVOL1* in the hair follicle cycle is rarely reported. Our study of OVOL1 expression changes during yak hair follicle cycle will provide clues to reveal the specific function of *OVOL1* in HFs development.

## 5. Conclusions

In summary, a lncRNA-mRNA co-expression network was constructed to identify the key pathways and the hub genes closely related to the development of yak HFs cycle. A series of well-known biological processes or pathways associated with HFs development were enriched by analyzing the temporally specific co-expression gene modules. The crucial genes in different stages of yak HFs cycle were screened by co-expression network analysis, combined with different molecular experimental methods. Our findings systematically elucidated the biological processes and important regulators during the development of yak HFs cycle, which would contribute to better understanding of molecular mechanism in the development of hair follicle and provide a useful reference information for molecular breeding in yak hair traits.

## Figures and Tables

**Figure 1 genes-13-00032-f001:**
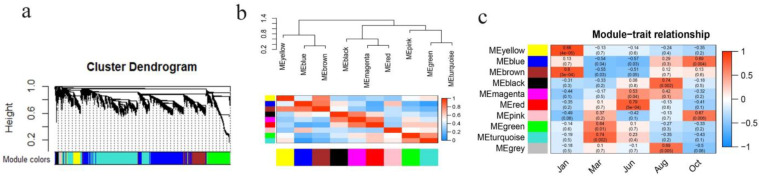
Construction of mRNA-lncRNA co-expression modules of yak HFs cycle based on DEMs and DELs. (**a**) Clustering dendrograms of co-expression network modules, each module was assigned a color. (**b**) The adjacency heatmap of eigengene. Below is the module clustering heatmap, above is the module clustering tree—high adjacency was shown in red, low adjacency was shown in blue. (**c**) Correlation analysis of modules and different periods of hair cycle. The upper number in the color block represents the correlation, below is the P value. Red indicates a positive correlation and blue indicates a negative correlation.

**Figure 2 genes-13-00032-f002:**
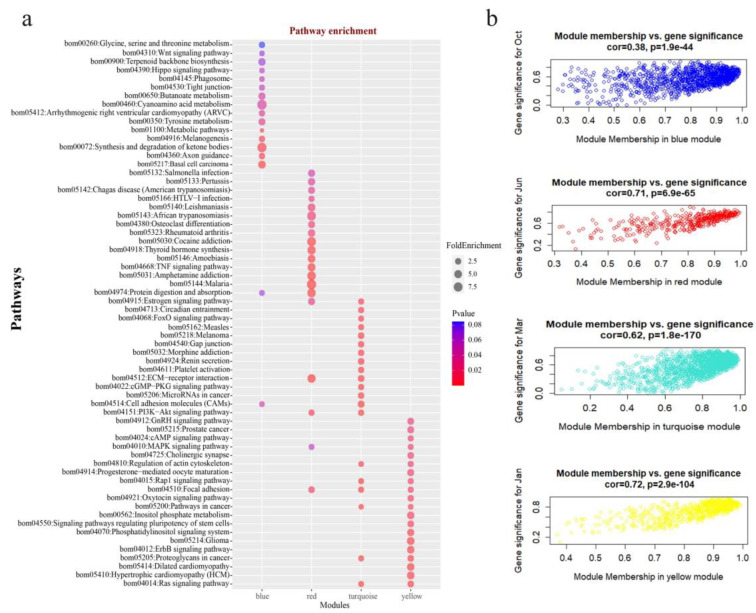
KEGG enrichment analysis of the key modules. (**a**) KEGG analysis of four temporally specifical modules including blue, red, turquoise, and yellow modules. The top 20 terms in different modules were visualized—if the enrichment result in a module were less than 20, all the terms would be showed. (**b**) The scatterplot of module membership (MM) vs. gene significance (GS) of the four modules in their associated periods, respectively.

**Figure 3 genes-13-00032-f003:**
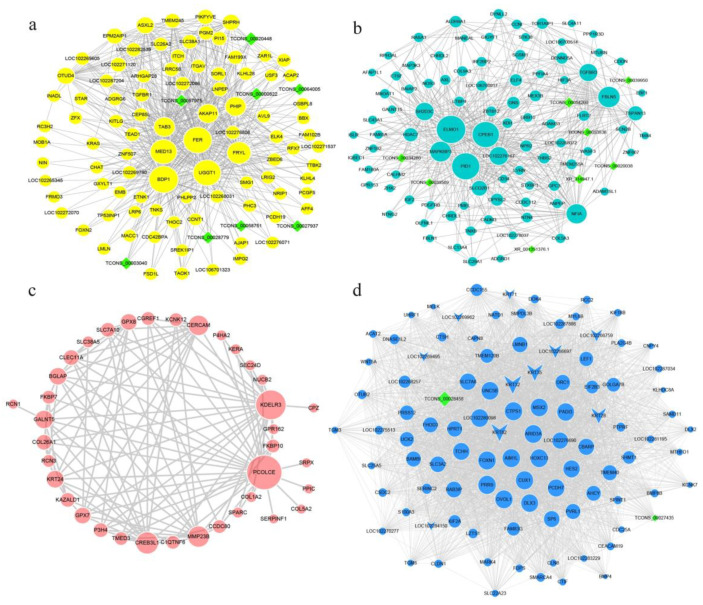
The co-expression network of the genes in top 100 connectivity in the key four modules. The networks are represented with their corresponding module colors, including the yellow, turquoise, red, and blue module networks (**a**–**d**). Genes in the red module network are less than 100 that were all exhibited. Circle represents mRNAs, diamond represents lncRNAs, the V shape in the blue network represents the keratins. The size of nodes represents the connectivity of genes in the co-expression network—the larger the nodes, the higher the connectivity and the more critical it is in the network and could be used as hub genes.

**Figure 4 genes-13-00032-f004:**
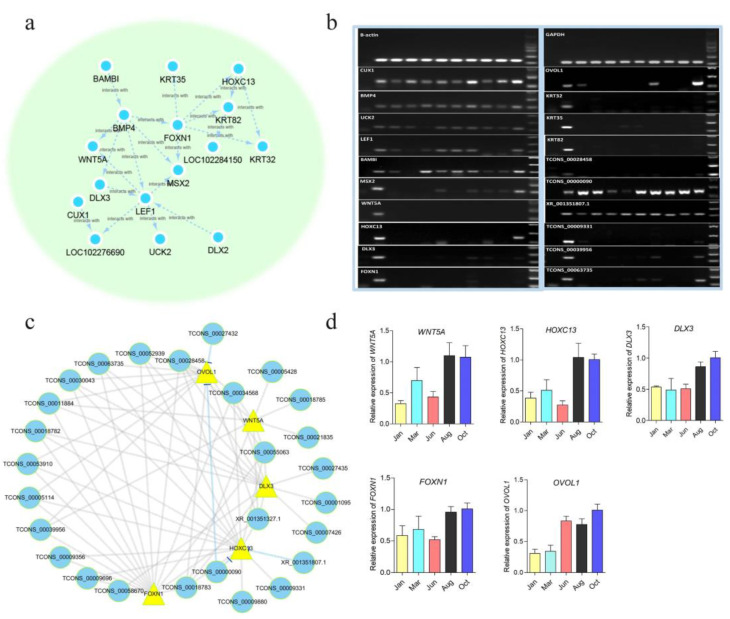
Analysis of PPI and expression characteristics of the hub genes in the blue module. (**a**) A subnetwork of PPI analysis result for the top 100 genes in the blue module. (**b**) PCR detected the tissue specific of a part of hub genes in the blue module network. The lanes from left to right represent the negative control, skin, small intestine, heart, liver, spleen, lungs, kidney, fat, muscle, testis, and 2000 Marker. (**c**) The interaction network of five mRNAs that specifically expressed in skin with its associated lncRNAs, which was analyzed by the Pearson correlation coefficient (Pearson correlation ≥0.90 or ≤0.90) and visualized by Cytoscape. The circles in blue represent lncRNAs and the triangles in yellow represent the mRNAs. The lines in blue and the arrows indicate negative effect, and others represent positive regulation. (**d**) RT-qPCR results for *WNT5A*, *HOXC13*, *DLX3*, *FOXN1*, and *OVOL1* during the HFs cycle of yak.

**Figure 5 genes-13-00032-f005:**
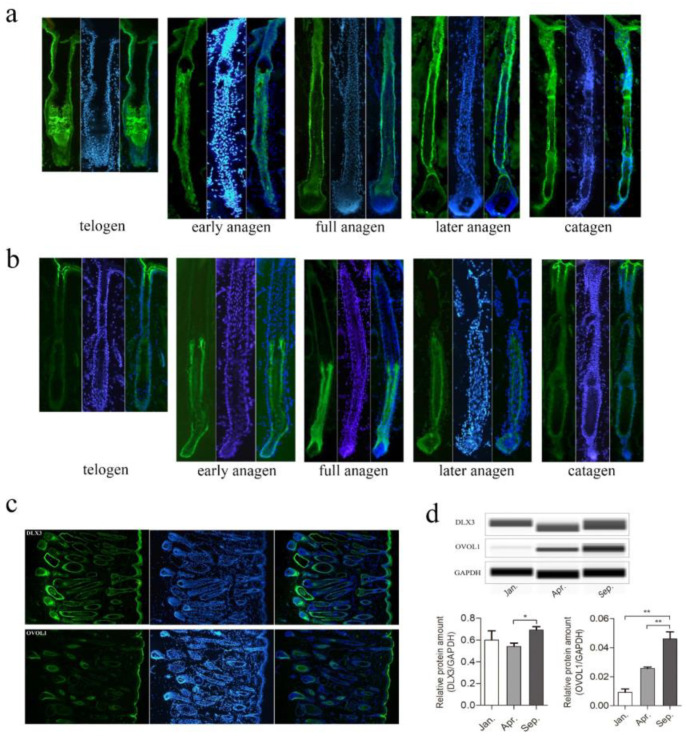
Detection of spatiotemporal expression of DLX3 and OVOL1 during yak HFs cycle by immunofluorescence. (**a**,**b**) The expressions of DLX3 (**a**) and OVOL1 (**b**) were detected using anti-Dlx3 and anti-Ovol1 antibody (green), respectively, in hair follicle at telogen, early anagen, full anagen, later anagen, and catagen. Staining in each period was represented by a single hair follicle—blue indicates DAPI staining. (**c**) Panoramic display of a microscope field in 10× of DLX3 (upper) and OVOL1 (lower) immunofluorescence staining. (**d**) Western blot analyses for DLX3 and OVOL1 protein levels in Jan. (catagen), Apr. (telogen), and Sep. (anagen), during yak HFs cycle, and the quantitative analysis of gray value were shown in histograms. Data are presented as mean ± SEM for 3 biological replicates; * *p* < 0.05; ** *p* < 0.01.

## Data Availability

The datasets during the current study available from the corresponding author on reasonable request, and the datasets analyzed during the current study are also available in the NCBI repository, accession number: PRJNA550233 (https://www.ncbi.nlm.nih.gov/bioproject/PRJNA550233, accessed on 20 October 2020).

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
