# Peer review of "Identification of the Key Genes Associated with the Yak Hair Follicle Cycle"

_genes, 2021, doi:10.3390/genes13010032_

Round 1

Reviewer 1 Report

Overall, this is an interesting subject that is still little discussed in the literature; thus, this article can contribute to better understanding of development of hair follicle in yak and contribute to the identification of key genes and important regulatory elements. However, the way the results were shown and discussed needs to be improved. Some points must be improved for the article. Please, see further comments.

In the material and methods, it is not possible to understand what was actually used as input to run the co-expression analyses. I understood that the authors used FPKM data from differentially expressed (DE) genes and lncRNAs from a published article, however, there is no information about comparison groups, for example under what conditions these transcripts were DE, comparing comparing follicular development in January with March, March with October and so on? which genes were on the list of genes used as input, were those combined results from different DE comparing sets, if so, why ? Would it provides a better explain from a biological point of view. Despite of set of mRNAs and lncRNAs used to performed the analysis showed in the present study, have been selected from a published article, does not remove the need for this article to have the basic information necessary for the reader to understand it without necessarily reading the other article, an article has to be self-explanatory, the reader cannot be dependent on another article to have a clarity about this article. It gives the impression that this is the functional analysis and validity part of a large experiment that the authors are trying to publish it separately.

The authors have to provide a list mRNA and lncRNAs used to perform the co-expression analysis, as supplementary table, and they have to report in which situation those transcripts were differentially expressed, in addition, the results and discussions sections should also be readjusted to include this information as needed (e.g., line 174).
The information in several figures isn’t clear (figure 2a; figure 3 and figure 4). The quality has to be improved or results must be displayed in a way that is readable to the reader. 
The discussion has to be improved in order to connect the ideas shown and discuss the results found, there are many sentences dropped without being properly discussed (e.g., line 374 to 384), which gives an idea of superficiality and it is not evident what this study brings of new and interesting.
It would be great to indicate in the discussion, which table or figure are being discussed (e.g., line 333, 337, 344). It is difficult to have to look for the figure that is being discussed; it is much easier if this information is indicated in the text.
The conclusion section needs to be rewritten; It does not report a conclusion. This sounds like a paragraph in the results section, it doesn't say what was concluded with the findings.

Reviewer 2 Report

The paper deals with the hair follicles in yak. It is well made and brings a lot of new knowledge. It needs minor revisions, mainly of formal character.

In Introduction, please distinguish clearly guard hair and undercoat.

English is on the whole good, but some sentences are too long and tangled:

r. 64 However, the regularly…

r. 230 In figure…

r. 266 It has been…

r. 288 To further… Too long, too tangled sentence!

r. 291 Results of…

r. 339 Mar. and Jun…….

Section 2.8 Statistical not statical.

Section 3.2, move the references into the Discussion section.

Rr. 359 and 368 twice point.

R. 378 several or serval.

Last paragraph, check the writing of gene abbreviations in italic, and proteins in normal type.
